# The Role of Symbiotic Microorganisms, Nutrient Uptake and Rhizosphere Bacterial Community in Response of Pea (*Pisum sativum* L.) Genotypes to Elevated Al Concentrations in Soil

**DOI:** 10.3390/plants9121801

**Published:** 2020-12-18

**Authors:** Andrey A. Belimov, Alexander I. Shaposhnikov, Darya S. Syrova, Arina A. Kichko, Polina V. Guro, Oleg S. Yuzikhin, Tatiana S. Azarova, Anna L. Sazanova, Edgar A. Sekste, Vladimir A. Litvinskiy, Vladimir V. Nosikov, Aleksey A. Zavalin, Evgeny E. Andronov, Vera I. Safronova

**Affiliations:** 1All-Russia Research Institute for Agricultural Microbiology, Podbelskogo sh. 3, Pushkin, 196608 Saint-Petersburg, Russia; ai-shaposhnikov@mail.ru (A.I.S.); syr_daria@ro.ru (D.S.S.); 2014arki@gmail.com (A.A.K.); polinaguro@gmail.com (P.V.G.); yuzikhin@gmail.com (O.S.Y.); tatjana-aza@yandex.ru (T.S.A.); anna_sazanova@mail.ru (A.L.S.); sekste_edgar@mail.ru (E.A.S.); eeandr@gmail.com (E.E.A.); v.safronova@rambler.ru (V.I.S.); 2Pryanishnikov Institute of Agrochemisty, Pryanishnikova str. 31A, 127434 Moscow, Russia; vl.litvinsky@gmail.com (V.A.L.); vniiasekr@yandex.ru (V.V.N.); otdzem@mail.ru (A.A.Z.); 3Department of Biology, Saint-Petersburg State University, University Embankment, 199034 Saint-Petersburg, Russia

**Keywords:** aluminium, mycorrhiza, nodulation, nutrient uptake, pea, PGPR, rhizosphere microbiome, soil acidity

## Abstract

Aluminium being one of the most abundant elements is very toxic for plants causing inhibition of nutrient uptake and productivity. The aim of this study was to evaluate the potential of microbial consortium consisting of arbuscular mycorrhizal fungus (AMF), rhizobia and PGPR for counteracting negative effects of Al toxicity on four pea genotypes differing in Al tolerance. Pea plants were grown in acid soil supplemented with AlCl_3_ (pH_KCl_ = 4.5) or neutralized with CaCO_3_ (pH_KCl_ = 6.2). Inoculation increased shoot and/or seed biomass of plants grown in Al-supplemented soil. Nodule number and biomass were about twice on roots of Al-treated genotypes after inoculation. Inoculation decreased concentrations of water-soluble Al in the rhizosphere of all genotypes grown in Al-supplemented soil by about 30%, improved N_2_ fixation and uptake of fertilizer ^15^N and nutrients from soil, and increased concentrations of water-soluble nutrients in the rhizosphere. The structure of rhizospheric microbial communities varied to a greater extent depending on the plant genotype, as compared to soil conditions and inoculation. Thus, this study highlights the important role of symbiotic microorganisms and the plant genotype in complex interactions between the components of the soil-microorganism-plant continuum subjected to Al toxicity.

## 1. Introduction

Elevated concentration of mobile aluminium ions is the main reason for phytotoxicity of acid soils resulting in the inhibition of plant growth and limitation of crop productivity [1,2,3]. The mechanisms of plant tolerance to Al toxicity have been intensively studied and involve exudation of organic acids and H^+^ ions from roots and secretion of mucilage to immobilize Al in the rhizosphere, internal detoxification within plant tissues, sequestration of Al in the vacuole, induction of antioxidative activity and efflux of Al from the root tissues [4,5,6,7,8,9]. These studies were performed mostly with wheat, barley, maize, soybean and *Arabidopsis thaliana* and demonstrated the prevalence of a particular mechanism for different plant species and cultivars. As for pea (*Pisum sativum* L.), differences in growth response to toxic Al between cultivars [10,11], the importance to counteract Al-induced oxidative stress [7,12,13], immobilization Al in roots by pectin [14] and the protective effect of micronutrient boron [15] were described. Our previous report demonstrated valuable intraspecific variability of pea in Al tolerance and showed that the increase in the rhizosphere pH, Al precipitation in root zone and maintenance of the plant nutrient homeostasis are principal tolerance mechanisms of this species [16].

Pea (*Pisum sativum* L.), being a legume species, may be considered as a relatively Al-sensitive crop as compared to cereals [11,17,18,19]. A vulnerability of leguminous plants in terms of Al toxicity is a high sensitivity in the formation of symbiosis with microorganisms, particularly with nitrogen-fixing nodule bacteria [20,21]. Negative effects of Al on nodule initiation, induction of oxidative stress in nodules and inhibition of nitrogen fixation were reported for pea [22,23,24]. At the same time Al-tolerant and efficient rhizobia nodulating various legume crops were characterized [25,26,27,28,29]. Moreover, *Rhizobium* sp. isolated from nodule of chick pea was able to bind Al^3+^ due to production of siderophores, suggesting capability of this bacterium to protect the plant against Al toxicity [30]. However, little is known about rhizobia forming efficient symbiosis with pea grown in acid soils and the role of such bacteria in combating Al stress in legume plants.

On the other hand, Al tolerant symbiotic arbuscular mycorrhizal fungi (AMF) are often present in acid soils and can alleviate toxicity of this element for plants [31,32]. The main mechanism of beneficial effect of AMF is related to mobilization of soil P, resulting in formation of insoluble phosphates with Al in the rhizosphere and inside plant roots. Another mechanism is due to the improved uptake of other nutrients (Ca, Mg, K and Fe) by plants, which is often inhibited by Al [33,34,35,36]. Such effects were described for several plant species but not for pea. It was also shown that many soil bacteria are tolerant to toxic Al concentrations due to efflux of Al from cells and exudation of Al-binding ligands [37,38]. However, interactions of such bacteria, including plant growth-promoting rhizobacteria (PGPR), with plants were scarcely studied. The Al tolerant PGPR strain *Viridibacillus arenosi* IHBB7171 produced auxins, possessed 1-aminocyclopropane-1-carboxylate (ACC) deaminase and stimulated growth of pea, but its effect on the plants under Al stress was not studied [39]. Inoculation of maize plants grown in acid soil with P-solubilizing *Burkholderia* sp. decreased Al accumulation in roots, promoted root elongation and thereby combated Al toxicity [40]. These findings suggest that symbiotic microorganisms may play important role in counteracting negative effects of Al on plants.

Increased root exudation of organic compounds in response to Al toxicity was repeatedly described for various plant species [2,3,5], including pea [16]. This can exert a significant effect on the composition and activity of rhizosphere microorganisms [41], since they use root exudates as a nutrient source and thereby interact with plants [42]. Genotype specific changes in the rhizosphere microbial community were also observed in soybean cultivars differing in Al tolerance [43,44]. However, up to now the role of rhizosphere microbiome in plant response to elevated Al concentrations in acid soils is little studied.

The aim of our study was to evaluate the potential of symbiotic microorganisms for improving growth and nutrient uptake of plants grown in acid soil and to estimate their role in adaptation of plants to Al toxicity. For this purpose, four pea genotypes differing in Al tolerance and a microbial consortium consisting of AMF, rhizobia and PGPR were used. An attempt was also made to relate the response of plants to Al and inoculation with changes in the composition of the rhizosphere microbial communities.

## 2. Results

### 2.1. Plant Biomass

Inoculation with microbial consortium increased shoot biomass of pea genotypes VIR1903 and VIR7307 grown in both neutralized and Al-supplemented soils (Figure 1a). Seed biomass increased after inoculation of VIR1903 and VIR8473 grown in Al-supplemented soil, as well VIR8473 in neutralized soil (Figure 1b). A positive effect of the introduced microorganisms on seeds of VIR8473 was mainly due to the increase in seed number per plant (Figure 1c). Biomass production and seed number were often less in Al-supplemented soil as compared with neutralized soil and this effect varied depending on pea genotype (Figure 1).

### 2.2. Symbiotic Structures

Nodule number (Figure 2a) and biomass (Figure 2b) on roots of VIR1903, VIR8353 and VIR8473 grown in neutralized soil was significantly (about two times) increased by inoculation. In Al-supplemented soil such effect on nodule number was evident for genotypes VIR1903, VIR7307 and VIR8473 (Figure 2a), on nodule biomass it was evident for genotypes VIR1903 and VIR8473 and a tendency for increased nodule biomass was observed for VIR8353 (*p* = 0.03; *n* = 4; Student’s *t* test) and VIR7307 (*p* = 0.005; *n* = 4; Student’s *t* test). Treatment with Al decreased nodule number of inoculated VIR8353 and uninoculated VIR8473 (Figure 2a) and nodule biomass of uninoculated VIR1903 and inoculated VIR8353 (Figure 2a,b). The inoculated genotype VIR8353 showed higher values of colonization intensity by mycorrhizal fungi and relative arbuscular richness in both soils as compared to uninoculated plants (Figure 2c,d). Relative arbuscular richness also increased in roots of the inoculated VIR7307 grown in neutralized soil (Figure 2d). Inoculation increased relative vesicular richness of VIR7307 and VIR8353 plants grown in Al-supplemented soil by three and four times, respectively (Figure 2e). Colonization of roots by *Ps. fluorescens* SPB2137 varied from 1 × 10^5^ to 7 × 10^5^ CFU g^−1^ root FW; however, no significant differences between treatments or genotypes were found (Appendix A).

### 2.3. Rhizosphere pH and Al Concentrations

In the end of experiment, all pea genotypes grown in neutralized soil had pH of the rhizosphere about 7.0, whereas pH was about 5.5 for the plants grown in Al-supplemented soil (Figure 3a). Inoculation had no effect on rhizosphere pH in both soils (Figure 3a). Rhizosphere Al concentration of all uninoculated genotypes grown in Al-supplemented soil was about twice as compared with neutralized soil (Figure 3b). Inoculation significantly decreased by about 30% rhizosphere Al concentration of all pea genotypes grown in Al-supplemented soil. All pea genotypes had increased Al concentrations in shoots when grown in Al-supplemented soil (Figure 3c). The exception was Al-treated and inoculated genotype VIR8473 showing decreased shoot Al concentration by 14% as compared with uninoculated plants (Figure 3c). Inoculation also tended to decrease shoot Al concentration in VIR8353 by 6% (*p* = 0.04; *n* = 4; Student’s *t* test). Seed Al concentrations were not affected by Al or inoculation with exception of inoculated genotype VIR8473 showing decrease by 21% and 33% (*p* = 0.04 and *p* = 0.001; *n* = 4; Fisher’s LSD test) when grown in neutralized and Al-supplemented soil, respectively (data not shown).

### 2.4. Nitrogen Uptake

Inoculation increased N concentration in shoots and seeds, as well as N content in shoots, of Al-treated VIR1903 (Table 1). Seed N content was also increased in the inoculated VIR8353 grown in neutralized soil. Positive effect of inoculation was observed on shoot ^15^N fraction and ^15^N content of VIR1903 in neutralized soil and shoot ^15^N fraction of VIR8473 in Al-supplemented soil (Table 1). Seeds of the inoculated Al-treated VIR8473 had increased ^15^N fraction and ^15^N content and VIR8353 seeds had increased ^15^N content (Table 1). On the other hand, inoculation decreased ^15^N fraction and ^15^N content in shoots and seeds of VIR8353 and VIR8473 grown in neutralized soil.

Inoculated plants VIR8353 and VIR8473 grown in neutralized soil, as well as VIR1903 and VIR8473 grown in Al-supplemented soil, had increased total N content (Figure 4a). Total ^15^N content increased in VIR1903 and VIR8473 grown in neutralized and Al-supplemented soil, respectively, but it decreased in VIR8353 grown in neutralized soil (Figure 4b). The effect of Al on N and/or ^15^N uptake by shoots and seeds varied depending on pea genotype and inoculation from positive (e.g., VIR7307) to negative (e.g., VIR8353) (Table 1). As a result, inoculated Al-treated plants VIR1903 had increased total N content, but the effect of Al on other inoculated pea genotypes was opposite (Figure 4a). Increased or decreased total ^15^N content was obtained for VIR7307 or VIR8353 and VIR8473 when the Al-treated plants were not inoculated (Figure 4b).

### 2.5. Phosphorus Uptake

Concentration of P in shoots of all pea genotypes grown in neutralized and/or Al-supplemented soils increased due to inoculation (Figure 5a). Positive effect of inoculation on seed P concentration was also observed in VIR 8353 (Figure 5b). Consequently, higher shoot (Figure 5c) and/or seed (Figure 5d) P contents were obtained in the inoculated genotypes VIR8353 and VIR8473 resulting in the increased total P content in plants grown in both soils (Figure 5e). Treatment with Al of inoculated VIR1903 and VIR8353, as well as uninoculated VIR7307, plants increased P concentration in shoots (Figure 5a). However, shoot, seed and/or total P contents of Al-treated plants was often less than that of plants grown in neutralized soil (Figure 5c–e).

### 2.6. Concentration of Other Nutrients in the Rhizosphere

Concentration of several water-soluble nutrients increased in the rhizosphere of inoculated plants, basically grown in Al-supplemented soil (see Figure 6 for details). Particularly for these plants, the most pronounced effects of inoculation were obtained for Fe (VIR1903), Mn (all pea genotypes), Mo (VIR1903, VIR7307 and VIR8473), Ni (VIR1903, VIR7307 and VIR8353), P (VIR1903, VIR7307 and VIR8473), S (VIR1903, VIR7307 and VIR8353) and Zn (VIR8353). Inoculation also increased rhizosphere concentration of Mo of VIR1903 and VIR8353 (Figure 6d), Ni of VIR8473 (Figure 6e), P of VIR8473 (Figure 6f), S of VIR1903 (Figure 6g) and Zn of VIR8353 (Figure 6h) grown in neutralized soil. A negative effect of inoculation was observed on Fe concentration of VIR7307 (Figure 6a), Mg concentration of VIR1903 (Figure 6b) and Zn concentration of VIR 1903 (Figure 6h) grown in Al-supplemented soil.

However, treatment with Al increased rhizosphere concentration of several nutrients, particularly Fe and Mn (Figure 6a,c), decreased rhizosphere concentration of Mo (Figure 6d). Differences in rhizosphere concentrations of Mg, Ni, P, S and Zn between plants grown in neutralized and Al-supplemented soils were also observed, but they varied from positive to negative depending on pea genotype and inoculation (Figure 6b,e–h). Concentrations of other nutrients (B, Ca, Co, Cu and K) were scarcely and/or in rare cases affected by soil conditions and pea genotypes (Appendix A).

### 2.7. Uptake of Other Nutrients by Plants

Uptake of N and P by plants has been described above in Section 2.4 and Section 2.5. Concentration of several nutrients in shoots is given in Figure 7. The inoculated VIR1903 showed increased shoot concentration of B (Appendix A), Cu (Appendix A), K (Figure 7b), Mo (Figure 7e) and Zn (Figure 7h) grown in neutralized soil, as well as of Cu (Appendix A) and K (Figure 7b) grown in Al-supplemented soil. The inoculated VIR7307 showed increased shoot concentration of Cu (Appendix A), K (Figure 7b) and Zn (Figure 7h) but decreased concentrations of Mo (Figure 7e), S (Figure 7g) grown in neutralized soil. Inoculation of VIR8353 decreased shoot concentration of Mn (Figure 7d), Mo (Figure 7e) and Zn (Figure 7h) grown in neutralized soil, as well as of Mn (Figure 7d) and Zn (Figure 7h) grown in Al-supplemented soil. However, Mo (Figure 7e) or S (Figure 7g) concentrations increased in the inoculated VIR8353 grown in Al-supplemented or neutralized soil, respectively. When VIR8473 was grown in neutralized soil, the inoculated plants had increased concentrations of all nutrients presented in Figure 7 and B, Co and Cu (Appendix A). Such positive effect was observed for Ca (Appendix A), Fe (Figure 7a), Mg (Figure 7c) and Ni (Figure 7f) in shoots of Al-treated VIR8473 plants.

Nutrient concentrations in seeds of VIR1903 and VIR7307 were not affected by inoculation, except increased B and Zn in seeds of VIR1903 and Ni in seeds of VIR7307 grown in neutralized soil (Appendix A). Seeds of inoculated VIR8353 had higher concentrations of all determined nutrients, except for Mn, Mo and Ni, when plants were grown in Al-supplemented soil. Such positive effect of inoculation on VIR8353 seeds was less pronounced in neutralized soil, but significant differences were observed for Co, Cu, Fe, K, Mg, P, S and Zn. Increased B, Ca, Co, Fe, K, Mg and S concentrations were found in seeds of inoculated VIR8473 grown in Al-supplemented soil (Appendix A). Seeds of VIR7307, VIR8353 and VIR8473 plants grown in Al-supplemented soils with or without inoculation usually had higher concentrations of various nutrients as compared with plants grown in neutralized soil (see Appendix A for details). An exception was Mo, the concentration of which decreased in seeds of all genotypes grown in Al-supplemented soil (Appendix A).

### 2.8. Rhizosphere Bacterial Communities

According to the taxonomic analysis, the total community consisted of 20 phyla with dominance of *Proteobacteria* (14–34%), *Actinobacteria* (17–28%), *Tahumarchaeota* (*Archaea*) (4–10%), *Verrucomicrobia* (3–10%) and *Acidobacteria* (5–7%). The less abundant phyla were *Firmicutes* (1–3%), *Planctomycetes* (1–2%), *Gemmatimonadetes* (0.4–2%) and *Bacteroidetes* (0.3–1%). The rest of the communities consisted of minor taxa and unassigned microorganisms (Figure 8).

In general, the rhizosphere taxonomic structure was relatively similar and scarcely affected by soil supplements and inoculation with microbial consortium but varied depending on plant genotype. Particularly, genotype VIR 8353 possessed relatively high proportion of genera *Stenotrophomonas*, *Burkholderia-Caballeronia-Paraburkholderia* and *Yesinia*.

The basic diversity indices including richness, evenness and phylogenetic metrics are given in the Appendix A. According to the data obtained there were no significant differences in diversity indices between treatments of different soil conditions, plant genotype and microbial inoculation. The UniFrac approach was used to study microbial communities’ separation trends related to the effects of Al, plant genotype and inoculation (Figure 9). The separation was more pronounced for the plant genotype and for the weighted UniFrac (3–5% against 28–33% of explained variance for unweighted and weighted UniFrac respectively).

The DESeq2 was used to identify phylotypes with statistically significant changes in their abundances and the fraction of microbiome affected by a particular factor and hereinafter referred to as “active fraction” (Figure 10a). For the plant genotype effect there were identified 27 differentially presented phylotypes making up 11.6% of the total community and belonging to *Proteobacteria*, *Verrucomicrobia*, *Actinobacteria* and *Archaea*. For factors Al and Al/Inoculation (Al + M) there were 18 and 9 phylotypes with 2.7% and 5.5% of total community abundance, respectively, whereas for factor Inoculation (M) there was not any detectable effect (Figure 10b).

## 3. Discussion

### 3.1. Plant Biomass

Our previous study with hydroponics showed that treatment with 80 µM AlCl_3_ for 10 days decreased root and shoot biomass of VIR1903 and VIR8473 about two times, whereas growth of VIR7307 and VIR8353 was not significantly affected [16]. It was the reason for taking these pea genotypes in the present study as Al-sensitive and Al-tolerant, respectively. However, the expected genotypic differences in growth response to Al toxicity were not found when the plants were cultivated in soil. Under soil conditions VIR1903 can be considered as more tolerant to Al, since only shoot biomass of uninoculated plants decreased in Al-supplemented soil (Figure 1a). The more Al-sensitive pea genotype was VIR8473, since Al treatment decreased shoot and seed biomass and seed number of both inoculated and uninoculated plants (Figure 1). In this respect, genotypes VIR7307 and VIR8353 occupy an intermediate position. The observed inconsistency in genotypic response to Al-toxicity might be due to different growth conditions in hydroponics and soil. Interestingly, VIR1903 exuded higher amounts of organic acids, particularly succinate, as compared to other studied genotypes cultivated in hydroponics [16]. The chelation of Al with organic acids exuded by roots is considered as one of the most important mechanisms of plant tolerance to this toxicant [2,3,5,8,9]. However, the amount of exuded organic acids did not correlate with growth inhibition by Al treatment of hydroponically cultivated pea genotypes [16]. Nevertheless, complexing of Al by these compounds might be important in soil system leading to the increase in Al tolerance of VIR1903.

The plants grown in Al-supplemented soil as a rule had less biomass and seed number, suggesting presence of stress caused by Al (Figure 1). Inoculation with microbial consortium increased shoot and/or seed biomass with variation depending on pea genotype. Namely, the inoculated VIR1903 had increased shoot biomass, genotype VIR8473 had increased seed biomass, whereas no effect was observed on VIR8353. A positive effect of inoculation was evident for both plants grown in neutralized and Al-supplemented soils. The results showed genotype dependent response of pea to the introduced microbes. It is in line with the report demonstrating very high polymorphism of pea in interactions with AMF and rhizobia [45,46]. Our results confirmed that inoculation with symbiotic microbes, such as AMF [31,32,33], rhizobia [30] or PGPR [40], improve plant growth in the presence of toxic Al in soil; however, this is the first time we have described such an effect on peas and applied a consortium of all three microsymbionts for this purpose.

### 3.2. Symbiotic Structures

The observed negative effect of Al on the number and/or biomass of pea nodules confirmed previous reports showing that nodulation process is sensitive to Al toxicity in various legumes [20,21], including pea [22,23,24]. Here, indigenous rhizobia presented and nodulated uninoculated pea plants grown in Al-supplemented soil and inoculation with *R. leguminosarum* bv. *viciae* RCAM1079 significantly increased nodule number and biomass. This suggests that the nodules were formed with Al-tolerant rhizobia. Previous reports described Al-tolerant *R. leguminosarum* bv. *trifolii* nodulating clover [25], *R. miluonense* [29] and *R. leguminosarum* bv. *phaseoli* nodulating common bean [27], *Bradyrhizobium* sp. nodulating mung bean [26], *Sinorhizobium meliloti* nodulating alfalfa [28] and *Rhizobium* sp. nodulating chick pea [30]. In our experiment, the nodule number positively correlated with nodule biomass (r = +0.76; *p* < 0.001; *n* = 64) and the latter positively correlated with shoot (r = +0.29; *p* = 0.019; *n* = 64) and seed (r = +0.50; *p* < 0.001; *n* = 64) biomass. This indicates an important role of nitrogen fixing symbiosis for pea growth and adaptation to elevated Al concentration on soil. 

The uninoculated pea plants were actively colonized by indigenous AMF in both neutralized and Al-supplemented soils (Figure 2). Most probably this was the reason for minor effect of the introduced *Glomus* sp. 1Fo on quantitative parameters of mycorrhizal infection. It was difficult to establish what proportion strain *Glomus* sp. 1Fo occupied. Nevertheless, positive effects of inoculation with *Glomus* sp. 1Fo on AMF colonization were observed in roots of VIR7307 and VIR8353. It is known that Al tolerant AMF present in acid soils and help plants to alleviate toxicity of this element [31,32]. Here, there was a positive correlation between relative vesicular richness and shoot biomass of VIR1903 (r = +0.54; *p* = 0.030; *n* = 16). Mycorrhizal colonization intensity in roots of VIR8353 positively correlated with shoot (r = +0.57; *p* = 0.025; *n* = 16) and seed (r = +0.64; *p* = 0.008; *n* = 16) biomass, and with seed number (r = +0.62; *p* = 0.010; *n* = 16). Similar positive correlations were found for the relative arbuscular richness in roots of this pea genotype (r varied from +0.60 to +0.64; *p* < 0.015; *n* = 16). In contrast, the relative arbuscular richness in roots of VIR7307 negatively correlated with seed biomass (r = −0.60; *p* = 0.015; *n* = 16). Moreover, mycorrhizal colonization intensity in roots of VIR8473 negatively correlated with seed biomass (r = −0.62; *p* = 0.010; *n* = 16) and seed number (r = −0.57; *p* = 0.022; *n* = 16), as well as the relative arbuscular richness of this genotype negatively correlated with seed biomass (r = −0.59; *p* = 0.016; *n* = 16). This observation showed that the intensity of the formation of mycorrhizal structures associated with opposite effects on plant growth and depended on the plant genotype. Interactions between AMF and plants are very complicated and varied from mutualism to parasitism [47,48,49,50]. For example, inoculation with a high inoculum density of *Glomus* spp. increased mycorrhization of roots but decreased root and shoot biomass of pea [51]. Xavier and Germida [52] did not find correlation between colonization intensity of roots by AMF and growth parameters of pea, and the effects of AMF on plant growth varied from positive to negative depending on fungal species. It was also shown that AMF induced retardation of pea development, which, however, did not lead to a decrease in plant productivity [53]. However, little is known about the specificity of the plant growth depression by AMF, which is due to the plant genotype within one species.

Behaviour of the introduced PGPR populations in the rhizosphere of plants subjected to Al stress was scarcely studied. We found no significant differences in the number of *Ps. fluorescens* SPB2137 in the rhizosphere of pea genotypes grown in both soils. This may suggest that *Ps. fluorescens* SPB2137 has high Al tolerance and shows a nonspecific interaction with the studied genotypes. Interestingly, inoculation with PGPR *Herbaspirillum seropedicae* increased nitrogen fixation and shoot N concentration only in Al-tolerant rice cultivars, which exuded bigger amounts of carbon into the rhizosphere, as compared to Al-sensitive cultivars [54]. However, the survival rate of *H. seropedicae* on roots was similar for all rice cultivars.

### 3.3. Rhizosphere pH and Al Concentrations

Rhizosphere pH was not affected by inoculation of pea plants grown in neutralized and Al-supplemented soils. This means that the observed decrease in mobile forms of Al in the rhizosphere of inoculated plants (Figure 3b) was not due to changes in soil pH. To explain the observed effect, it can be assumed that the introduced microorganisms released into the rhizosphere various substances that bound aluminium into insoluble forms. Probably, this was a defence response of microorganisms, since the effect manifested itself only at an increased concentration of Al in the rhizosphere when adding this toxicant. All components of the microbial consortium could take a part in this phenomenon, since the ability to bind Al and alleviate Al toxicity was previously described for AMF [31,32], nodule bacteria [30] and PGPR [40]. Another possibility is that the introduced microorganisms activated exudation of Al-binding compounds by pea roots. It was shown earlier that inoculation with *Ps. putida* increased exudation of organic compounds of wheat and maize by about two times [55]. An increase in root exudation can occur due to an additional concentration gradient of organic compounds directed from the root and caused by the trophic activity of microorganisms [56], as well as due to the influence of microbial metabolites [57,58]. The ability of PGPR to increase the intensity of photosynthesis [59,60] also can increase the influx of photosyntates into the roots and thereby activate exudation.

Aluminium concentration in the rhizosphere positively correlated with Al concentration in pea shoots (r = +0.42; *p* < 0.001; *n* = 64) but negatively correlated with seed number (r = −0.33; *p* = 0.007; *n* = 64). This indicated the interrelation of rhizosphere processes with Al uptake by shoot and plant productivity. In turn, shoot Al concentration negatively correlated with shoot (r = −0.76; *p* < 0.0001; *n* = 64) and seed (r = −0.54; *p* < 0.0001; *n* = 64) biomass, seed number (r = −0.72; *p* < 0.0001; *n* = 64), nodule number (r = −0.27; *p* = 0.029; *n* = 64) and nodule biomass (r = −0.50; *p* < 0.0001; *n* = 64). These correlations point to the toxic effect of Al absorbed by plants on pea growth and symbiosis with nodule bacteria and complement previous reports about Al toxicity for pea [7,10,11,12,13,16] and its nodulation efficiency [20,21,22,23,24]. Our study for the first time connected Al availability in the rhizosphere and Al uptake by plant shoots with immobilization of this toxicant by the introduced symbiotic microorganisms.

### 3.4. Nitrogen Uptake

The pea genotypes grown in Al-supplemented soil had similar (VIR8353), decreased (VIR8473) or even increased (VIR1903 and VIR7307) shoot N concentrations, and VIR1903 had increased seed N concentration, as compared with those grown in neutralized soil (Table 1). This suggests that Al did not induce N deficiency and the observed decrease in shoot and seed N contents (Table 1, Figure 4a) in such plants was due to inhibition of plant growth in Al-supplemented soil. Opposite effects of these soils on the uptake of ^15^N also evident and depended on pea genotype. Such a complex situation may be a consequence of the simultaneous participation of genotypic differences in Al tolerance, efficiency of interaction with rhizobia and assimilation of mineral N. High intraspecies variability of pea in these traits was described previously [16,45]. In addition, shoot Al concentration positively correlated with shoot (r = +0.78; *p* < 0.0001; *n* = 64) and seed (r = +0.46; *p* < 0.001; *n* = 64) N concentration, but negatively correlated with shoot (r = −0.32; *p* = 0.010; *n* = 64) and seed (r = −0.42; *p* = 0.001) N content. The rhizosphere Al concentration negatively correlated with shoot N content (r = −0.25; *p* = 0.043; *n* = 64) and ^15^N content (r = −0.38; *p* = 0.002; *n* = 64). It was shown previously that Al decreased shoot biomass and N concentration of pea grown in Al-supplemented hydroponics as a result of inhibition of N-fixing symbiosis with rhizobia [22]. Here for the first time we showed that the uptake of N and fertilizer ^15^N by pea plants grown in Al-supplemented soil was generally less as compared with those grown in neutralized soil (Figure 4). At the same time, inoculation with the microbial consortium alleviated negative effects of Al on nitrogen nutrition. The observed effects might be due to both the improved N_2_ fixation by symbiosis with *R. leguminosarum* bv. *viciae* RCAM1079 and increased fertilizer or soil N uptake in symbiosis with *Glomus* sp. 1Fo and *Ps. fluorescens* SPB2137.

### 3.5. Phosphorus Uptake

In line with the results obtained for N (see above), the Al-treated pea plants had increased shoot P concentrations and decreased shoot P contents (Figure 5). Inoculation positively affected both P concentration and P content in shoots and seeds, confirming that the improvement of P uptake and subsequent immobilization of Al with phosphates in mycelium and plant tissues are important mechanisms for counteraction of Al toxicity by AMF [31,32,61]. Increased shoot P concentrations in Al-treated plants due to inoculation with AMF were shown for barley [35], sorghum [34], broomsedge [62] and tulip-poplar [36]. Here we expanded it for pea plants and also showed genotype dependent effect.

In our experiment, shoot Al and P concentration did not correlate (data not shown). However, a negative correlation was found between Al and P concentrations in the rhizosphere of Al-treated plants (r = −0.56; *p* = 0.001; *n* = 32). The ability of AMF to decrease Al concentration in the sand, where tulip-poplar plants were cultivated, was previously shown [36]. Our results suggest that immobilization of Al in the rhizosphere with phosphates might occur and contribute to alleviation of Al toxicity for pea roots. On the other hand, the observed effect could be due to P mobilization activity not only by AMF *Glomus* sp. 1Fo but also by *Ps. fluorescens* SPB2137, since this strain (unpublished data) and this PGPR species actively solubilize both inorganic and organic phosphates [63,64].

### 3.6. Uptake of Other Nutrients by Plants

A lower concentration of several nutrients (particularly Fe, Mg and Mn) in the rhizosphere of pea plants grown neutralized soil as compared with Al-supplemented soil was probably due to differences in soil pH values. It is known that mobility of these nutrients is closely depended on soil pH and usually higher in acid soils; however, mobility of Mo, P and S in acid soils is low [65,66]. These assumptions are in line with the obtained results for nutrient concentrations in the rhizosphere of the studied peas (with variations depending on genotype) when comparing plants grown in two soils (Figure 5 and Figure 6; Appendix A). Our original observation is that inoculation increased nutrient concentrations in the rhizosphere, particularly counteracting the decrease in Mo, P and S concentrations in the rhizosphere of Al-treated plants. Molybdenum is an important element for nitrogen fixation being a cofactor of nitrogenase [67], whereas P [68,69] and S [70] are crucial elements for Al tolerance in plants.

On the other hand, relatively high availability of several nutrients, such as Fe, K, Mg, Mn and Ni (Figure 6; Appendix A), in Al-supplemented soil might neutralize negative effect of Al on plant mineral nutrition. Previously we showed significant inhibition of nutrient uptake caused by Al toxicity in hydroponically grown peas, including the studied genotypes [16]. It was concluded that maintenance of nutrient homeostasis is a crucial Al tolerance mechanism for pea. However, here we speculate that the importance of this mechanism could be limited in acid soil, where availability of many nutrients was relatively higher as compared to neutralized soil. Taking this into account, the reason for the discrepancy between the Al tolerance of the studied pea genotypes in hydroponics [16] and soil (this study) might be the different availability of nutrients. In addition, the physicochemical and environmental differences between hydroponics and soil, as well as resident soil microorganisms, could influence interactions of pea plants with Al.

Positive effects of AMF on shoot concentrations of nutrients, such as Ca, Fe, Mg and Mn, in plants subjected to Al stress ware reported for switchgrass [33], sorghum [34], barley [35,71] and tulip-poplar [36]. Our study showed significant increase in shoot and seed concentrations of these and other nutrients in pea grown in Al-supplemented soil and inoculated with microbial consortium containing AMF *Glomus* sp. 1Fo (Figure 7, Appendix A). It can be assumed that this was mainly due to the interaction of peas with *Glomus* sp. 1Fo. This hypothesis is supported by the increase in nutrient uptake by pea grown in Cd-supplemented soil and inoculated with a consortium, in which this AMF strain was used with other rhizobia and PGPR strains [72]. Influence of inoculation with rhizobia and/or PGPR on the concentration of these nutrients in plants in the presence of toxic Al received little attention in the literature.

### 3.7. Rhizosphere Bacterial Communities

The observed taxonomic structures have common traits with the earlier described wheat and pea rhizospheric microbiomes but differ from those by shifts in abundance of few taxonomic groups only [73,74,75,76], e.g., representatives of *Proteobacteria*, *Firmicutes* and *Thaumarchaeota*. Analysis of the alpha-diversity showed no significant differences in diversity indices between treatments with different soil conditions, plant genotype and microbial inoculation. This might be explained by the fact that the studied factors affected only small fraction of the total microbiome.

Beta diversity patterns suggest that the main factor shaping the taxonomic structure of the rhizosphere microbiome in this experiment is the plant genotype as compared to soil conditions and inoculation. The mechanism of this separation can be explained by variation of taxa abundances in rhizosphere of different plant genotypes rather than in taxa presence/absence patterns, that was evident from the difference between separations in weighted in unweighted modes. This assumption was analysed in detail with statistics of differential abundances of a particular phylotype. The most notable differences observed in rhizosphere microbiomes caused by plant genotype were related to the differential abundance of *Alpharoteobacteria* (*Stenotrophomonas*), *Gammaproteobacteria* (*Burkholderia-Caballeronia-Paraburkholderia, Bradyrhizobium, Yersinia*, Unclassified *Burkholderiaceae*) and *Actinobacteria* (*Gaiella*). It is known that many representatives of *Burkholderia*, *Enterobacter* and *Bradyrhizobium* are PGPR and have the capacity to solubilize phosphates [77,78]. However, in our experiment there were no clear interrelations between the abundance of that bacteria and concentration of mobile phosphorus in the rhizosphere.

To compare effects of the studied factors (plant genotype, soil conditions and inoculation) on the rhizosphere community structure we calculated the “active fraction” of the microbiome characterizing a sum of all phylotypes with statistically significant changes in their abundances in response to the particular factor. The size of this fraction indicated how big the part of the total microbiome was affected by each factor and their interactions. Accordingly to the size of the “active fraction”, we ranged factors in their significance: plant genotype > aluminium > aluminium + inoculation > inoculation. It is in line with observations where plant genotype was an important factor for shaping rhizosphere microbiome in the presence of toxic Al [43,44], but AMF inoculation had no effect on the alpha diversity [79] and PGPR had minor effect on the rhizosphere microbiome [80]

## 4. Materials and Methods

### 4.1. Plants and Microorganisms

Seeds samples of four pea (*Pisum sativum* L.) genotypes VIR1903, VIR7307, VIR8353 and VIR8473 263 were obtained from the N.I. Vavilov Institute of Plant Genetic Resources (Saint-Petersburg, Russia) and multiplied by authors of the manuscript. Nodule bacterium *Rhizobium leguminosarum* bv. *viciae* RCAM1079 [81], AMF *Glomus* sp. 1Fo [82] and rifampicin resistant (20 mg L^−1^) variant of PGPR *Pseudomonas fluorescens* SPB2137 [83] were obtained from the Russian Collection of Agricultural Microorganisms (RCAM, St.-Petersburg, Russia, http://www.arriam.ru/kollekciya-kul-tur1/). During the experimental work, the bacteria were maintained on agar yeast extract mannitol (YM) agar [84]. Inoculum *Glomus* sp. 1Fo was obtained by growing mycorrhized plants of Swedish ivy (*Plectranthus australis* L.) in sterilized soil and preparing a mixture of soil and roots with a total intensity of mycorrhizal infection of 93%. Similar soil-root mixture containing no endomycorrhizal fungi was used for inoculation as a control treatment.

### 4.2. Plant Growth Conditions

The used sod-podzolic light loamy soil had the following characteristics determined by standard methods (mg kg^−1^): total C, 26,200 ± 650; total N, 1900 ± 110; nitrate N, 15 ± 2; ammonium N, 23 ± 4; available P, 35 ± 6; available K, 59 ± 9; water soluble Al, 15 ± 3. The soil also had: total exchangeable bases, 64 ± 5 mg equiv; exchangeable and specifically bound forms of Al (extract 1 M HNO_3_), 660 ± 45; hydrolytic acidity, 5.8 ± 0.7 mmol kg^−1^; pH_KCl_ = 4.5 ± 0.1; pH_H2O_ = 5.2 ± 0.2. One half of soil was additionally enriched with AlCl_3_ at final concentration of 50 mg Al kg^−1^. Another half of soil was supplemented with CaCO_3_, yielding pH_KCl_ = 6.2 ± 0.3 and pH_H2O_ = 7.0 ± 0.3, and named here as neutralized soil. Pots were filled with 2 kg soil and each pot was fertilized with nutrient solutions resulting in (mg kg^−1^): KCl, 300; MgSO_4_, 30; CaCl_2_, 20; H_3_BO_3_, 3; MnSO_4_, 3; ZnSO_4_, 3; Na_2_MoO_4_, 2. Nitrogen fertilizer was added as ^15^NH_4_^15^NO_3_ in the amount of 45 mg kg^−1^ with a final enrichment by 35 atom% ^15^N. The soil was supplemented (or not) with CaCO_3_, yielding pH_KCl_ = 6.2 ± 0.2. Then the soil was watered to 80% of water holding capacity (WHC) and incubated at room temperature for 5 days for stabilization.

A pot experiment was carried out in a polyethylene greenhouse with natural lighting and temperature in summer (June–August, 2016, St.-Petersburg). The average monthly temperature, humidity and daylight hours were for June +17.4 °C, 65.4%, 18 h, for July +19.9 °C, 67.8%, 18 h and for August +18.4 °C, 70.1%, 17 h, respectively (Gismeteo, https://www.gismeteo.ru/). Pea seeds were selected for homogeneity of seed weight, then surface-sterilized and scarified by treatment with 98% H_2_SO_4_ for 10 min, rinsed with sterile water and germinated on moistened filter paper (Whatman #1) in the dark at 25 °C for 3 days. Each seedling was inoculated with a mixture of microorganisms: 1 mL of *R. leguminosarum* bv. *viciae* RCAM1079 and 1 mL of *Ps. fluorescens* SPB2137 water suspensions containing 10^7^ cells mL^−1^ each and supplemented with 5 g of mycorrhizal inoculum immediately after sowing. Control seeds were treated with 2 g of similar root-soil mixture containing no endomycorrhizal fungi. Four pots with 5 uniform seedlings per pot were prepared for each plant genotype and treatment. Pots were watered up to 70% WHC with evapotranspirational losses replenished every day by weighing the pots. The plants were cultivated for 80 days until maturity and harvested.

### 4.3. Symbiotic Parameters

At the end of experiment the roots were removed from 4 remaining pots and the soil attached to the root surface (rhizosphere soil) was collected by uniformly shaking the roots of each pot in sterile plastic bags. Each sample was divided into two parts. One part was frozen at −80 °C to determine the rhizosphere microbiome, and the other part was dried and used to determine the content of water-soluble forms of Al and nutrients.

Portion of roots about 100 mg fresh weight (FW) from each pot was taken to determine root colonization efficiency of *Ps. fluorescens* SPB2137 using its rifampicin resistance and the root homogenate dilution technique that has been recently described [72]. The number of *Ps. fluorescens* SPB2137 was expressed as colony forming units (CFUs) per g of root fresh weight (FW). The remaining roots were washed, and nodules were collected, counted, dried and weighed. Then the roots were prepared for estimation of mycorrhizal colonization as described by Turnau et al. [85] with some modifications. The root portions were treated with 10% KOH for 10 min at 95 °C and washed with water 3 times. The mycelium in the roots was stained by treatment for 3 min in 10% acetic acid supplemented with 5% black mascara solution (Sheaffer Pen, Shelton, CT, USA) and thoroughly washed with water. Microscopy of roots was performed using light microscope Axio Lab.A1 (Carl Zeiss, Oberkochen, Germany). Frequency of mycorrhizal structures (F), colonization intensity within mycorrhizal roots (M), relative arbuscular richness (A), and relative vesicular richness (V) were assessed as described previously [86]. Dry weight (DW) of roots and shoots from each pot (*n* = 4 per treatment and pea genotype) was determined.

### 4.4. DNA Extraction from Soil

DNA was extracted from 0.2 g of soil using the PowerSoil DNA Isolation Kit (Mobio Laboratories, Solana Beach, CA, USA), which included a bead-beating step, according to the manufacturer’s specifications. Samples were homogenized with a Precellys 24 (Bertin Corp., Rockville, MD, USA) at 6.5 m/sec, twice for 30 s. The purity and quantity of DNA were tested by electrophoresis in 1% agarose in 0.5× TAE buffer. The purified DNA templates were amplified with universal primers F515 5′-GTGCCAGCMGCCGCGGTAA-3′ and R806 5′-GGACTACVSGGGTATCTAAT-3′ [87] targeting the variable region V4 of bacterial and archaeal 16S rRNA genes, flanking an approximately 300-bp fragment of the gene, extended with service sequences containing linkers and barcodes according to Illumina technology. The PCR reactions were assembled in a 15 µL mix containing 1 u of Phusion Hot Start II High-Fidelity polymerase и 1X Phusion buffer (Thermo Fisher Scientific, Waltham, MA, USA), 5 pM of both primers, 1–10 ng of DNA, and 2 nM of each dNTP (Life Technologies, Carlsbad, CA, USA). The PCR thermal profile used was 94 °C for 30”, 50 °C for 30”, and 72 °C for 30” for 29 cycles. A final extension was performed at 72 °C for 3’. PCR products were purified and size selected with AM Pure XP (Beckman Coulter, Brea, CA, USA). Further library preparation was done according to the manufacturer’s protocol with the MiSeq Reagent Kit Preparation Guide (Illumina, San Diego, CA, USA). Libraries were sequenced on an Illumina MiSeq with an MiSeq^®^ Reagent Kit v3 (2 × 300b) sequencing kit.

### 4.5. Sequencing Data Processing

Raw sequence reads were deposited in SRA (NCBI) archive under BioProject ID PRJNA663278 within the dataset SUB8135271 (https://www.ncbi.nlm.nih.gov/Traces/study/?acc = PRJNA663278). Amplicon libraries of the 16S rRNA gene were processed using packages in the R [88] and QIIME2 [89] software environment. Rstudio [90] was used as the development environment for R. Raw sequence reads were trimmed and grouped into amplicon sequence variants (ASV, phylotypes) by use of dada2 package [91]. Classifier from package DECIPHER [92] trained on Silva 132 [93] was used to classify phylotypes. The phylogenetic tree was built in the QIIME2 software environment in the SEPP package [49]. Data was normalized by rarefaction algorithm according to the sample with the smallest number of readings (14000) for alpha and beta-diversity analysis. For alpha-diversity estimation Chao1, Observed phylotypes, Shannon, Simpson and Faith PD coefficients was computed, beta-diversity was estimated by unweighted- and weighted UniFrac metrics [94] in QIIME2 package. For differential analysis of phylotypes and quantitative metrics, the normalization was performed by variance stabilization algorithm through DESeq2 package [95]. To estimate the significance of differences between phylotypes, previously normalized data were processed using the Wald test, with Benjamin-Hotchberg false discovery rate (FDR) correction in the DESeq2 package. At the first stage we tested the significance factors in the total dataset and showed that the only significant factor was the plant genotype. To get more precise pictures we tested this significance for each particular plant genotype. In this mode we succeeded to get statistical support for the Al and Al + inoculation effect for some plant genotypes. Mean Log2Fold-Change was computed as average Log2Fold-Change for subset of differentially represented phylotypes characteristic for each group by factor: plant genotype, aluminium, inoculation, aluminium + inoculation interaction. The part of “active fractions” in the total community abundance was computed as summary abundance of all the phylotypes “active” for factor considered in total community. The R packages phyloseq [96], ggplot2 [97] and ggtree [98] were used for postprocessing and visualization of the obtained data.

### 4.6. Aluminium and Nutrient Contents

The dried plant shoots (leaves, stems and pod walls) and seeds were ground to a powder. Nitrogen content and atom% ^15^N in the ground samples were determined using the elemental analyzer (FlashEA 1112, Thermo Scientific, Italy) coupled with the isotope ratio mass spectrometer Delta V Advantage (Thermo Scientific, Dreieich, Germany) and the continuous flow interface ConFlo III following manufacturer’s instructions. To determine Al and nutrient (B, Ca, Co, Cu, Fe, K, Mg, Mn, Mo, Ni, S, P and Zn) contents, the ground shoot samples were digested in a mixture of concentrated HNO_3_ and 38% H_2_O_2_ at 70 °C using DigiBlock digester (LabTech, Sorisole, Italy). Rhizosphere soil samples (80 DAP) were incubated in deionized water (5 g soil + 25 mL water) at continuous shaking for 1 h. Soil samples were centrifuged at 10,000 g for 10 min and supernatants tested for pH_H2O_ using pH meter F20 (Mettler-Toledo, Schwerzenbach, Switzerland). Then soil supernatants were acidified with concentrated HNO_3_ up to 1% to suppress microbial activity. Elemental content of digested plant samples and soil supernatants were determined using an inductively coupled plasma emission spectrometer ICPE-9000 (Shimadzu, Kyoto, Japan) following manufacturer’s instructions.

### 4.7. Statistical Analysis

Statistical analysis of the data was performed using the software STATISTICA version 10 (TIBCO Software Inc., CA, USA). MANOVA analysis with Fisher’s LSD test and Student’s *t* test were used to evaluate differences between means.

## 5. Conclusions

Thus, inoculation of pea with the studied microbial consortium consisting of AMF, rhizobia and PGPR, increased biomass production and improved nutrition of plants grown in acid soil having elevated Al concentration, resulting in partial amelioration of Al toxicity. Pea genotypes significantly differed in Al tolerance and in response to the introduced microbes; however, no clear relationship between these traits was found. Positive effects of inoculation on Al-treated plants were accompanied by the increase in root mycorrhization and formation of symbiotic nodules, suggesting active interactions of plants with the introduced AMF and rhizobia. Our important and original observation was the decrease in concentration of mobile (water soluble) Al accompanied by the increase in macro- and micronutrients in the rhizosphere of inoculated plants. Thanks to this, we connected Al immobilization by the introduced symbiotic microorganisms in the rhizosphere with the uptake of this toxicant by plant shoots. We propose that the introduced microorganisms, particularly AMF *Glomus* sp. 1Fo, immobilized Al through the solubilization of soil phosphates and whereby mitigating Al toxicity. The increase in nutrient concentrations in the rhizosphere, particularly Mo, P and S, concentrated negative effect of Al on their mobility and uptake by Al-treated plants. Previously we showed significant inhibition of nutrient uptake caused by Al toxicity in the studied pea genotypes grown in hydroponics [16]. Although previously we postulated that maintenance of nutrient homeostasis is a crucial Al tolerance mechanism for pea in hydroponics, the importance of this mechanism could be limited in acid soil, where availability of many nutrients was relatively higher as compared to neutralized soil. The most prominent factor shaping the rhizosphere microbial community was the plant genotype highlighting it important role in complex interactions between the components of the soil-microorganism-plant continuum subjected to Al toxicity. However, the role of the observed phenomenon in plant tolerance to Al toxicity needs more detailed investigation. Thus, inoculation of pea with symbiotic microbial consortium increased Al tolerance of plants most probably due to immobilization of Al in the rhizosphere, decrease in Al concentrations in plants and improvement of N-fixation and nutrient uptake from soil.

## Figures and Tables

**Figure 1 plants-09-01801-f001:**
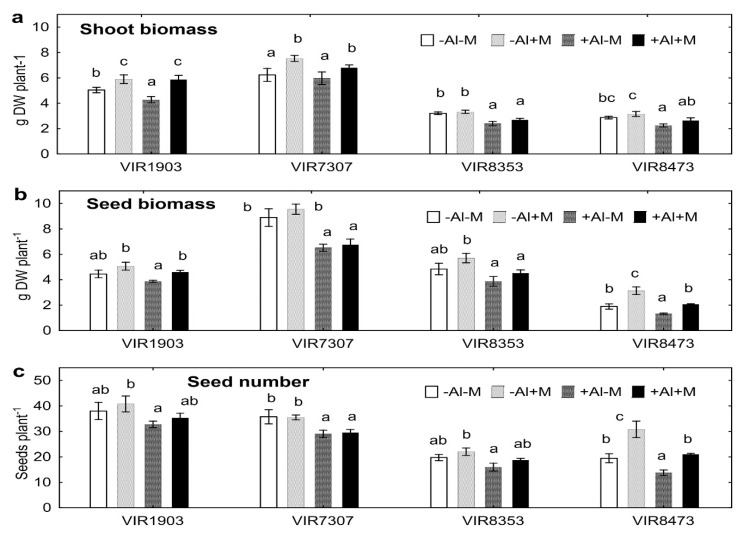
Shoot (**a**) and seeds (**b**) biomass and seed number (**c**) of pea genotypes VIR1903, VIR7307, VIR8353 and VIR8473 inoculated with microbial consortium and grown in neutralized or Al-supplemented soil. Treatments: -Al-M—neutralized soil with uninoculated plants, -Al + M—neutralized soil with inoculated plants, +Al-M—Al-supplemented soil with uninoculated plants, +Al + M—Al-supplemented soil with inoculated plants. Plants were inoculated with a microbial consortium consisting of *Pseudomonas fluorescens* SPB2137, *Rhizobium leguminosarum* bv. *viciae* RCAM1079 and *Glomus* sp. 1Fo. Vertical bars show standard errors. Different letters show significant differences between treatments within particular pea genotype (least significant difference test, *p* < 0.05, *n* = 4). DW stands for dry weight. Plants were analyzed on the 80th day after planting.

**Figure 2 plants-09-01801-f002:**
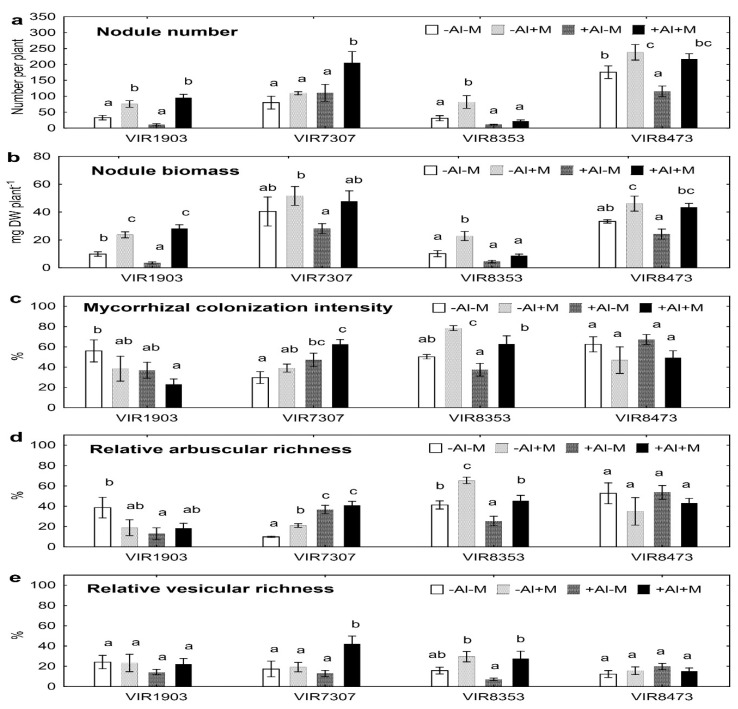
Nodule number (**a**), nodule biomass (**b**), root colonization intensity by mycorrhizal fungi (**c**), relative arbuscular richness (**d**) and relative vesicular richness (**e**) of pea genotypes VIR1903, VIR7307, VIR8353 and VIR8473 inoculated with microbial consortium. See details in the legend to Figure 1.

**Figure 3 plants-09-01801-f003:**
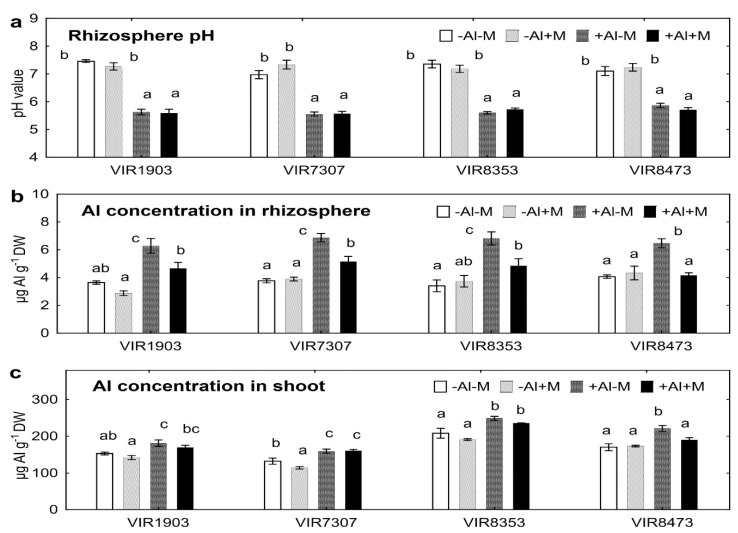
Rhizosphere pH (**a**) and concentration of Al in the rhizosphere (**b**) and shoots (**c**) of pea genotypes VIR1903, VIR7307, VIR8353 and VIR8473 inoculated with microbial consortium. See details in the legend to Figure 1.

**Figure 4 plants-09-01801-f004:**
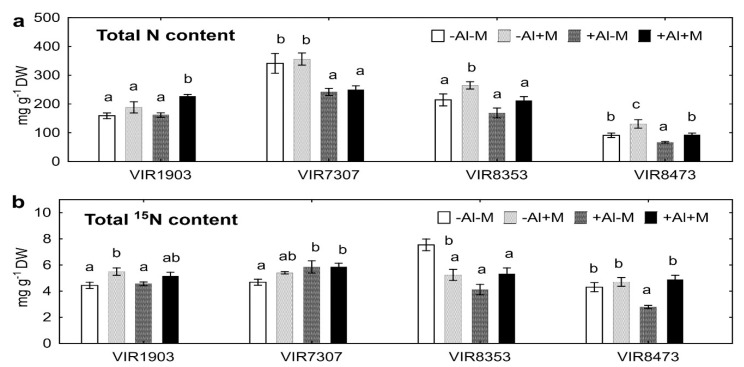
Total content (the sum of shoots and seeds) of N (**a**) and ^15^N (**b**) of pea genotypes VIR1903, VIR7307, VIR8353 and VIR8473 inoculated with microbial consortium. See details in the legend to Figure 1.

**Figure 5 plants-09-01801-f005:**
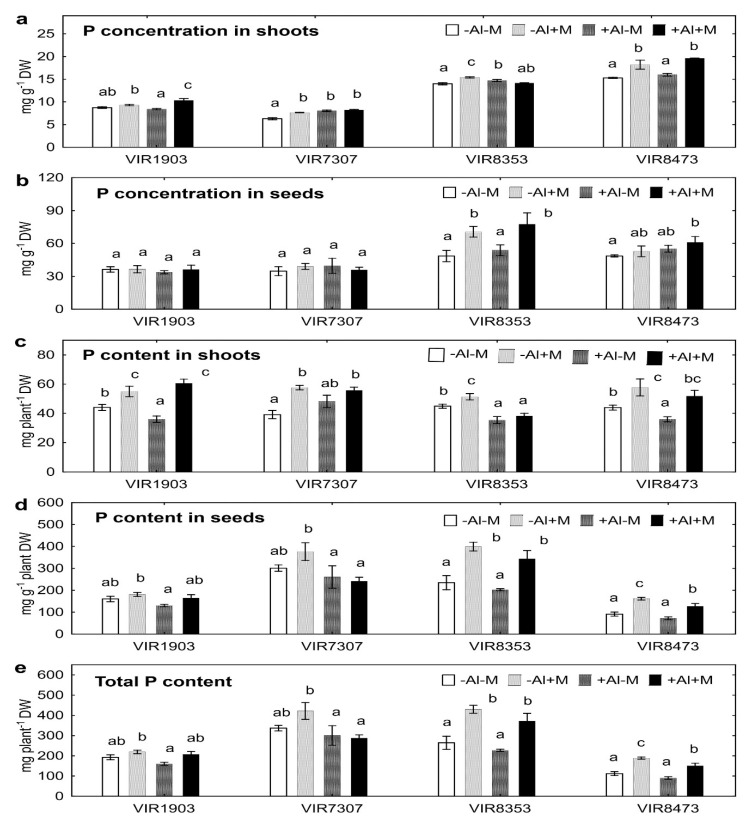
Phosphorus concentration in shoots (**a**) and seeds (**b**), P content in shoots (**c**) and seeds (**d**) and total P content (**e**) of pea genotypes VIR1903, VIR7307, VIR8353 and VIR8473 inoculated with microbial consortium. See details in the legend to Figure 1.

**Figure 6 plants-09-01801-f006:**
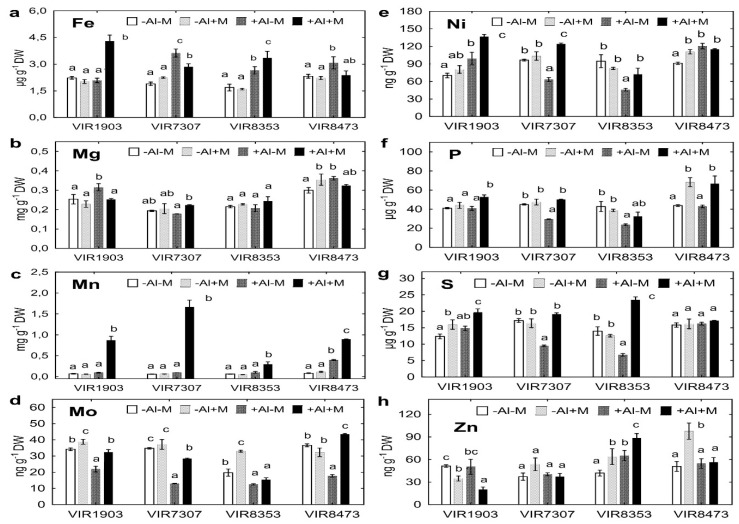
Concentration of nutrients Fe (**a**), Mg (**b**), Mn (**c**), Mo (**d**), Ni (**e**), P (**f**), S (**g**) and Zn (**h**) in the rhizosphere of pea genotypes VIR1903, VIR7307, VIR8353 and VIR8473 inoculated with microbial consortium. See details in the legend to Figure 1.

**Figure 7 plants-09-01801-f007:**
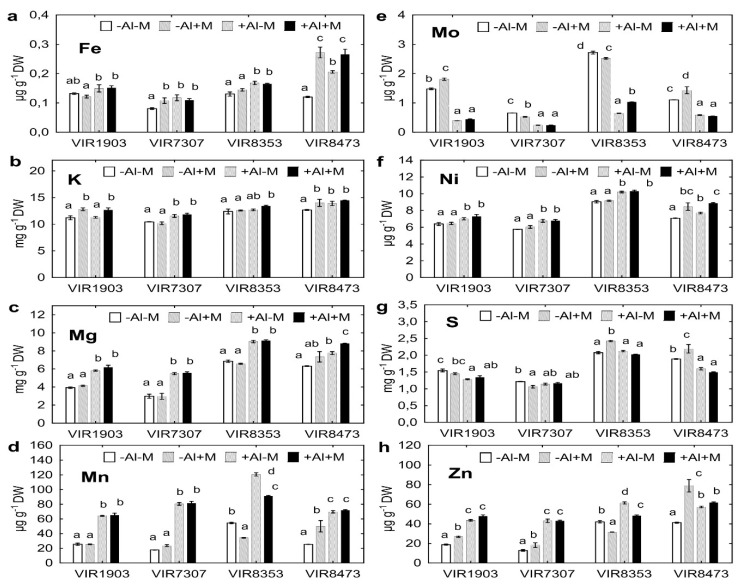
Concentration of nutrients Fe (**a**), K (**b**), Mg (**c**), Mn (**d**), Mo (**e**), Ni (**f**), S (**g**) and Zn (**h**) in shoots of pea genotypes VIR1903, VIR7307, VIR8353 and VIR8473 inoculated with microbial consortium. See details in the legend to Figure 1.

**Figure 8 plants-09-01801-f008:**
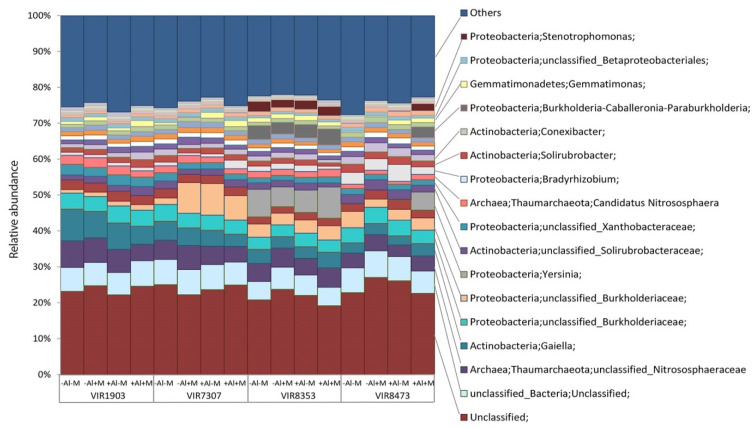
Procaryotic microbiome taxonomic structure at the genus level of the rhizosphere of pea genotypes VIR1903, VIR7307, VIR8353 and VIR8473 inoculated with microbial consortium. Others mean the sum of taxa with total abundance below 0.1%. See details in the legend to Figure 1.

**Figure 9 plants-09-01801-f009:**
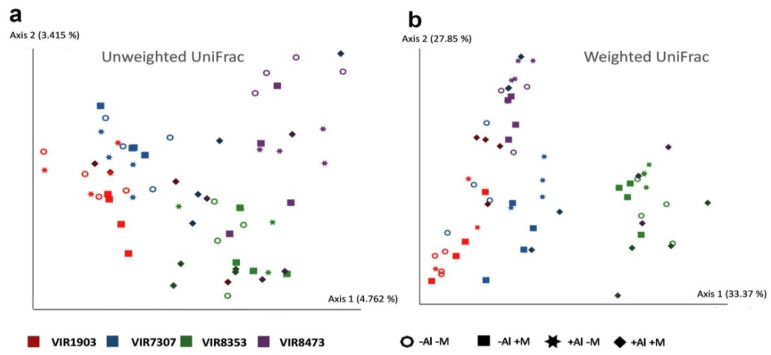
Beta-diversity plots evaluated by Unweighted (**a**) and Weighted (**b**) UniFrac for the rhizosphere prokaryotic microbiomes of pea genotypes VIR1903, VIR7307, VIR8353 and VIR8473 inoculated with microbial consortium. The parenthesized values are percent of variation described by each of axis. See details in the legend to Figure 1.

**Figure 10 plants-09-01801-f010:**
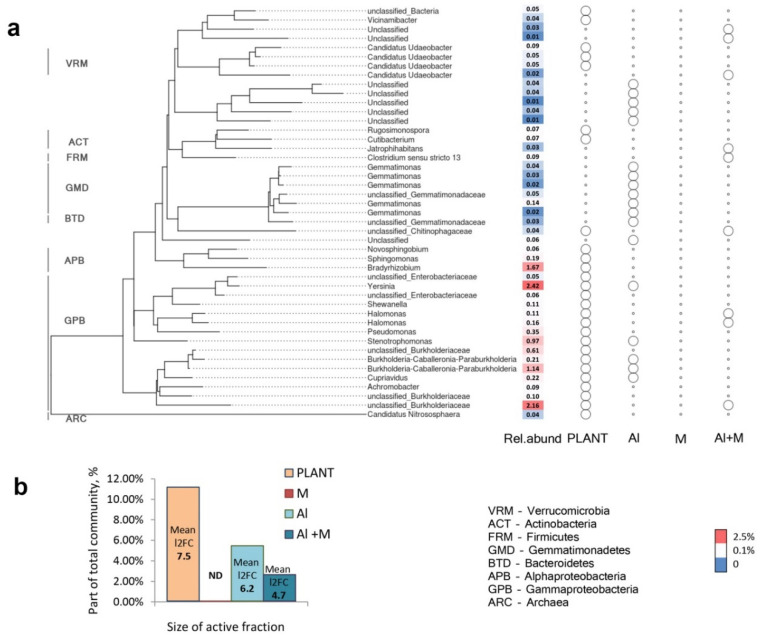
Phylogenetic tree of an “Active” microbiome fraction (**a**) and the size of an “Active” fraction as the percent of the total community affected by the particular factor (**b**). Factors: PLANT—pea genotype, Al—aluminium supplement of soil, M—inoculation with microbial consortium, Al + M—interaction between aluminium supplement of soil and inoculation with microbial consortium. On the right side of the tree the presence (big empty cycle) or the absence (small empty cycle) of each phylotype in an “Active” microbiome fraction for each factor are shown. ND stands for not detected. Column “Rel.abund” shows phylotype’s relative abundance in total community (see scale bottom right). l2FC means the magnitude of shift in response to each factor as an averaged log2-fold change value for all “active” phylotypes.

**Table 1 plants-09-01801-t001:** Concentration and content of nitrogen in shoots and seeds of pea genotypes inoculated with microbial consortium and grown in neutralized or Al-supplemented soil.

Treatments	Shoots				Seeds			
	N Concentration (mg g^−1^ DW)	^15^N Fraction (%)	N Content (mg plant^−1^)	^15^N Content (mg plant^−1^)	N Concentration (mg g^−1^ DW)	^15^N Fraction (%)	N Content (mg plant^−1^)	^15^N Content (mg plant^−1^)
Pea genotype VIR1903								
-Al −M	7.1 ± 0.2 ^a^	3.50 ± 0.01 ^c^	36 ± 1 ^a^	1.25 ± 0.03 ^b^	27.9 ± 2.0 ^a^	2.60 ± 0.16 ^a^	123 ± 11 ^a^	3.19 ± 0.27 ^a^
-Al +M	7.6 ± 0.1 ^a^	**4.44 ± 0.02 ^d^**	**45 ± 3 ^b^**	**1.98 ± 0.13 ^d^**	28.1 ± 1.3 ^a^	2.52 ± 0.21 ^a^	143 ± 17 ^ab^	3.51 ± 0.20 ^a^
+Al −M	7.9 ± 0.1 ^a^	3.25 ± 0.01 ^b^	34 ± 2 ^a^	1.11 ± 0.07 ^a^	33.0 ± 1.6 ^b^	2.72 ± 0.18 ^a^	128 ± 7 ^ab^	3.46 ± 0.20 ^a^
+Al +M	**10.0 ± 0.2 ^b^**	*2.79 ± 0.03 ^a^*	**59 ± 3 ^c^**	**1.64 ± 0.08 ^c^**	**36.8 ± 1.8 ^c^**	2.10 ± 0.20 ^a^	168 ± 6 ^b^	3.53 ± 0.35 ^a^
Pea genotype VIR7307								
-Al −M	6.4 ± 0.1 ^ab^	2.02 ± 0.01 ^a^	40 ± 3 ^a^	0.81 ± 0.06 ^a^	33.6 ± 1.4 ^b^	1.31 ± 0.09 ^a^	301 ± 33 ^b^	3.87 ± 0.19 ^a^
-Al +M	5.6 ± 0.2 ^a^	2.00 ± 0.01 ^a^	42 ± 1 ^ab^	0.84 ± 0.03 ^a^	32.8 ± 0.9 ^ab^	1.48 ± 0.12 ^a^	314 ± 21 ^b^	4.56 ± 0.09 ^a^
+Al −M	7.1 ± 0.2 ^b^	2.61 ± 0.01 ^b^	43 ± 4 ^ab^	1.12 ± 0.12 ^b^	30.5 ± 1.1 ^ab^	2.38 ± 0.23 ^b^	199 ± 13 ^a^	4.73 ± 0.46 ^a^
+Al +M	7.1 ± 0.1 ^b^	2.57 ± 0.01 ^b^	48 ± 2 ^b^	1.24 ± 0.06 ^b^	29.8 ± 0.6 ^a^	2.34 ± 0.24 ^b^	201 ± 15 ^a^	4.62 ± 0.28 ^a^
Pea genotype VIR8353								
-Al −M	13.4 ± 0.1 ^a^	5.60 ± 0.01 ^d^	43 ± 1 ^b^	2.41 ± 0.07 ^b^	35.2 ± 0.8 ^a^	3.08 ± 0.34 ^b^	171 ± 20 ^a^	5.13 ± 0.43 ^c^
-Al +M	13.3 ± 0.1 ^a^	*2.72 ± 0.01 ^a^*	44 ± 2 ^b^	*1.21 ± 0.05 ^a^*	38.7 ± 0.9 ^a^	*1.82 ± 0.09 ^a^*	**220 ± 11 ^b^**	*4.04 ± 0.37 ^b^*
+Al −M	13.1 ± 0.3 ^a^	3.36 ± 0.01 ^c^	32 ± 3 ^a^	1.06 ± 0.09 ^a^	35.5 ± 1.2 ^a^	2.28 ± 0.25 ^a^	137 ± 15 ^a^	3.07 ± 0.35 ^a^
+Al +M	13.8 ± 0.2 ^a^	*3.12 ± 0.01 ^b^*	37 ± 2 ^ab^	1.16 ± 0.05 ^a^	38.7 ± 0.6 ^a^	2.39 ± 0.20 ^a^	175 ± 12 ^a^	**4.17 ± 0.44 ^b^**
Pea genotype VIR8473								
-Al −M	12.4 ± 0.2 ^b^	4.17 ± 0.01 ^b^	36 ± 2 ^b^	1.49 ± 0.08 ^b^	29.4 ± 2.0 ^a^	5.10 ± 0.17 ^b^	55 ± 6 ^ab^	2.81 ± 0.29 ^b^
-Al +M	10.9 ± 0.7 ^ab^	*3.42 ± 0.03 ^a^*	34 ± 4 ^b^	*1.10 ± 0.15 ^a^*	30.4 ± 0.8 ^a^	*3.98 ± 0.60 ^a^*	96 ± 13 ^b^	3.61 ± 0.27 ^bc^
+Al −M	10.3 ± 0.2 ^a^	4.36 ± 0.01 ^c^	23 ± 1 ^a^	1.01 ± 0.04 ^a^	33.0 ± 0.9 ^a^	4.12 ± 0.34 ^a^	44 ± 3 ^a^	1.78 ± 0.14 ^a^
+Al +M	9.8 ± 0.9 ^a^	**4.46 ± 0.01 ^d^**	26 ± 2 ^a^	1.14 ± 0.10 ^a^	32.5 ± 2.0 ^a^	**5.54 ± 0.16 ^b^**	68 ± 4 ^ab^	**3.74 ± 0.25 ^c^**

Plants were inoculated with a microbial consortium consisting of *Pseudomonas fluorescens* SPB2137, *Rhizobium leguminosarum* bv. *viciae* RCAM1079 and *Glomus* sp. 1Fo. Different superscript letters (a, b, c and d) show significant differences between treatments within a subcolumn of particular pea genotype (least significant difference test, *p* < 0.05, *n* = 4). Data are means ± SE. Values in bold or italicized indicate significant positive or negative effects of the microbial consortium, respectively. DW stands for dry weight. Plants were analyzed on the 80th day after planting.

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
