# Peer review of "The Role of Symbiotic Microorganisms, Nutrient Uptake and Rhizosphere Bacterial Community in Response of Pea (*Pisum sativum* L.) Genotypes to Elevated Al Concentrations in Soil"

_plants, 2020, doi:10.3390/plants9121801_

Round 1

Reviewer 1 Report

The concept of this research is the need of the hour. Studying genotypes of crop plants is of importance in understanding plant-microbe interactions. Use of a small consortium of microbes also has significance in the current PMI research.  

From the manuscript presentation point of view - the main text needs some careful English language check, both grammatical and sentence construction. Also, some data has not been added as mentioned in the text, which actually strengthens the manuscript. 

For example -

Line 59 – Correct spelling of ‘Elevated’

Line 61 – should be ‘mechanisms of plant tolerance’

Line 78 – Sentence to be reconstructed.

Line 136 – This data should be included. Adds strength to the research.

Line 138 – This data should be included in the supplementary table. This also adds some meaning to the research.

Line 353 – correct as 11.6%

Line 557 – It can also be because of the resident microbes in the soil used in the experiment prior to adding the inoculum. Hydroponics is stressful to legumes too. A couple of references addressing these will help.

Figures -

Figures have some formatting issues such as, the alphabets indicating significance are not aligned well and some of the headings are not clear. Please have these corrected.

In all the Figure legends the treatments are mentioned in detail. The authors can mention this in the text in the methods section and shorten the legends.

Supplementary data Tables -

Table S3: The footnote mentions CFU, but this is not anywhere in the Table. Either justify this or remove it.

Table S4: The footnote below the table mentions the treatments in detail. It will be better if this note is moved up before Table 1 since all the tables have these treatments mentioned.

Author Response

Dear Reviewer,

Thank you very much for the valuable advice and comments. We tried to give answers to all questions to address all your comments.

Yours sincerely,

Andrey Belimov

Reviewer 1

Comments and Suggestions for Authors

The concept of this research is the need of the hour. Studying genotypes of crop plants is of importance in understanding plant-microbe interactions. Use of a small consortium of microbes also has significance in the current PMI research. 

From the manuscript presentation point of view - the main text needs some careful English language check, both grammatical and sentence construction. Also, some data has not been added as mentioned in the text, which actually strengthens the manuscript.

Response:

We carefully checked the manuscript and tried to improve English language and to correct mistakes. The changes are on lines: 79, 156, 236,238, 241, 243, 263, 311, 317, 334, 391, 476, 478-479, 457, 485, 511, 515, 547, 468, 577, 600, 605, 613, 641, 650, 783, 812. The required data are added (see below).

For example -

Line 59 – Correct spelling of ‘Elevated’

Response: The word “Elevated” has been corrected (Line 76).

Line 61 – should be ‘mechanisms of plant tolerance’

Response: The word “tolerance” has been corrected (Line 78).

Line 78 – Sentence to be reconstructed.

Response: The word “Although” is replaced by the words “At the same time” (Lines 94-95).

Line 136 – This data should be included. Adds strength to the research.

Response: The data on relative vesicular richness has been added to Figure 2 and the words “data not shown” are replaced by the words “Figure 2e” (Line 159).

Line 138 – This data should be included in the supplementary table. This also adds some meaning to the research.

Response: The data on colonization of roots by Ps. fluorescens SPB2137 has been added to the supplementary materials (Figure S1) and the words “data not shown” are replaced by the words “Figure S1” (Line 161). See also lines 824-826.

Line 353 – correct as 11.6%

Response: The correction has been made (Line 392).

Line 557 – It can also be because of the resident microbes in the soil used in the experiment prior to adding the inoculum. Hydroponics is stressful to legumes too. A couple of references addressing these will help.

Response: A sentence describing such possibilities has been added (Lines 605-607). We ask you to allow no references to the literature on these issues, because this is quite speculative.

Figures -

Figures have some formatting issues such as, the alphabets indicating significance are not aligned well and some of the headings are not clear. Please have these corrected.

Response: The alignment of alphabets indicating significance in the figures 1-7 has been improved. Sorry, it is not clear for me what headings need to be corrected. One correction is made in heading to Figure 4 (Line 247).

In all the Figure legends the treatments are mentioned in detail. The authors can mention this in the text in the methods section and shorten the legends.

Response: We ask you to allow no shorten the legends of figures. We tried to make the legends self-explanatory for each figure without reference to the Materials and Methods section. However, we understand your comment that the figure captions are fairly similar.

Supplementary data Tables -

Table S3: The footnote mentions CFU, but this is not anywhere in the Table. Either justify this or remove it.

Response: The CFU is removed from footnote of Table S3.

Table S4: The footnote below the table mentions the treatments in detail. It will be better if this note is moved up before Table 1 since all the tables have these treatments mentioned.

Response: We ask you to allow no removing the footnotes of these tables. We tried to make each table self-explanatory supplying a detailed footnote for avoiding looking for information on the file.

Reviewer 2 Report

Dear Authors,

The paper is an interesting case study. It provides a detailed analysis of the role of symbiotic microorganisms.
In my opinion this work should make a good contribution to the literature. Further more, in my opinion, the paper will attract a wide readership. The MS looks good. Therefore, I have no serious substantive comments to the MS. The experimental design and data analysis are appropriate, the introduction and discussion are consistent.
However, you'd need to think about the paper volume of work, it is in my opinion too long
My suggestions and minor comments:
1. Please improve the quality of figs. 1,2,3,5
2. Please read the MS once more and correct any minor shortcomings, e.g. punctuation, etc.

Author Response

Dear Reviewer,

Thank you very much for the valuable advice and comments. We tried to give answers to all questions to address all your comments.

Yours sincerely,

Andrey Belimov

Reviewer 2

Comments and Suggestions for Authors

Dear Authors,

The paper is an interesting case study. It provides a detailed analysis of the role of symbiotic microorganisms.

In my opinion this work should make a good contribution to the literature. Further more, in my opinion, the paper will attract a wide readership. The MS looks good. Therefore, I have no serious substantive comments to the MS. The experimental design and data analysis are appropriate, the introduction and discussion are consistent.

However, you'd need to think about the paper volume of work, it is in my opinion too long.

Response: Thank you very much for your positive attitude to our article! We agree with your comment that the article is voluminous. With this in mind we have tried to be concise in describing the results and many results are included as supplemental data. The large volume of the article is largely associated with a very voluminous experiment: 4 pea genotypes, 2 types of soil and 2 types of inoculation and many parameters measured. But, it is this that made it possible to identify interesting patterns. We also counted on the fact that according to the rules of the Biomolecules there is no restrictions on the length of research manuscripts.

My suggestions and minor comments:

  1. Please improve the quality of figs. 1,2,3,5

Response: We improved the alignment of alphabets indicating significance in the figures 1-7.

  1. Please read the MS once more and correct any minor shortcomings, e.g. punctuation, etc.

Response: We carefully checked the manuscript and tried to improve English language and to correct mistakes. The changes are on lines: 79, 156, 236,238, 241, 243, 263, 311, 317, 334, 391, 476, 478-479, 457, 485, 511, 515, 547, 468, 577, 600, 605, 613, 641, 650, 783, 812.